# On the Definition and Computation of Causal Treewidth

**Yizuo Chen**[1]                    **Adnan Darwiche**[1]

[1]Computer Science Department, University of California, Los Angeles, USA

## Abstract

Causal treewidth is a recently introduced notion allowing one to speed up Bayesian network inference and to bound its complexity in the presence of functional dependencies (causal mechanisms) whose identities are unknown. Causal treewidth is no greater than treewidth and can be bounded even when treewidth is unbounded. The utility of causal treewidth has been illustrated recently in the context of causal inference and model-based supervised learning. However, the current definition of causal treewidth is descriptive rather than perspective, therefore limiting its full exploitation in a practical setting. We provide an extensive study of causal treewidth in this paper which moves us closer to realizing the full computational potential of this notion both theoretically and practically.

## 1 INTRODUCTION

Treewidth is one of the most influential notions for parameterizing the complexity of probabilistic inference. This notion originated in the graph theory literature and can be viewed as a measure of graph connectivity [Robertson and Seymour, 1986]. It has also been used to parameterize the complexity of many algorithmic tasks that transcend probabilistic inference; see, e.g., [Bodlaender, 2006, Dechter, 2003]. For Bayesian networks, the time and space complexity of computing marginals is bounded by $O(n \cdot \exp(w))$ where $n$ is the number of nodes in the network and $w$ is its treewidth. For example, tree-structured networks have a treewidth $\leq 1$ so treewidth allows us to show that inference on such networks can be done in linear time and space.

Treewidth captures the structural aspects of a model and is independent of its parameters. Hence, one can use treewidth to provide guarantees on the complexity of inference without needing to know the model parameters. In the first few decades of research on Bayesian network inference, the perception was that high treewidth is a barrier since all influential algorithms at that time, particularly the jointree and variable elimination algorithms [Jensen et al., 1990, Zhang and Poole, 1996, Dechter, 1996], had a complexity which was also lower bounded exponentially by treewidth. Later developments showed that exploiting the parametric structure of Bayesian networks can lead to tractable inference in some situations where the treewidth can be very high; see, e.g., [Larkin and Dechter, 2003, Chavira and Darwiche, 2005, Chavira et al., 2006, Chavira and Darwiche, 2008]. The parametric structure exploited was particularly in the form of context-specific independence [Boutilier et al., 1996] and logical constrains (i.e., parameters in $\{0, 1\}$).[1]

More recently, a new and more abstract type of parametric structure has been identified and exploited computationally: functional dependencies, also known as causal mechanisms, which identities are *unknown* [Darwiche, 2020]. In a Bayesian network, a node is functionally determined by its parents if fixing the state of these parents also fixes the state of the node (that is, the node distribution is deterministic given any state of its parents). We often know that a node is functionally determined by its parents but without knowing the identity of the underlying function. This is prominent, for example, in causal inference where one typically has a causal graph in which every internal node is assumed to be functionally determined by its parents yet without knowing the specific functions that relate nodes to their parents [Pearl, 2000]. Classical techniques for exploiting parametric structure are not applicable in this case since

---

[1]Among the most effective approaches for exploiting parametric structure are the ones based on compiling Bayesian networks into tractable circuits [Darwiche, 2003]. These approaches allow one to conduct inference in time linear in the circuit size while yielding circuits whose size is not necessarily exponential in treewidth—see [Darwiche, 2021a] for a recent survey on circuit representations and [Agrawal et al., 2021] for a recent empirical evaluation in which methods based on circuits ranked at the forefront in terms of efficiency.

*Accepted for the 38th Conference on Uncertainty in Artificial Intelligence* (UAI 2022).

these methods require knowledge of the specific model parameters which imply knowledge of the specific functions that determine the values of internal nodes. This also arises when learning the parameters of a Bayesian network from data where we may have background knowledge to the effect that some nodes are functionally determined by their parents but without knowing the specific functions as we are trying to learn them; see, e.g., [Chen et al., 2020]. Interestingly enough, a recent finding showed that one can exploit unknown causal mechanisms computationally, leading to potentially exponential reduction in complexity [Darwiche, 2020]. This finding was based on two new theorems and cast in the context of model-based supervised learning. It particularly took the form of an algorithm that compiles the *structure* of a Bayesian network into a tractable circuit whose size is not necessarily exponential in treewidth. This approach managed to efficiently compile circuits for networks with treewidth over 100 without needing to know the network parameters, only that some nodes are functionally determined by their parents. More recently, this finding was cast in the context of causal inference while hinting that it can lead to a new parameter for bounding complexity that was called *causal treewidth* [Darwiche, 2021b].

Treewidth is classically defined for an undirected graph but it can be extended to directed acyclic graphs (DAGs) by computing the treewidth of the *moralized* DAG. This is an undirected graph obtained from the DAG by connecting every pair of parents by an edge and then removing the directionality of edges; see, e.g., [Darwiche, 2009, Ch 9]. Causal treewidth applies only to DAGs in which some nodes are declared as being *functional*. If no nodes are functional, then the causal treewidth reduces to treewidth. While [Darwiche, 2021b] suggested this more refined notion of causal treewidth, it did not provide an operational definition of causal treewidth and therefore it did not specify a method for computing it. Moreover, while [Darwiche, 2020] showed that inference can be sped up, exponentially in some cases, by exploiting unknown causal mechanisms, it did not fully exploit the two new theorems that enabled these techniques.

Our goal in this paper is to first review the two key theorems in [Darwiche, 2020] that enabled the computational exploitation of unknown causal mechanisms, and to then use them as basis for formally defining the notion of causal treewidth and how it can be computed. In the process of doing so, we will prove some results about the algorithmic techniques proposed in [Darwiche, 2020], showing that some are optimal while others are not. In other words, we will show that the algorithmic techniques proposed in [Darwiche, 2020] do not fully exploit the two enabling theorems identified in that work. Hence, the main contribution of this work is that it brings us closer, both theoretically and practically, towards the full exploitation of unknown causal mechanisms during inference. At a more cognitive level, our contribution may provide further hints as to why causal knowledge is so

| $A$ | $B$ | $C$ | $f_C(ABC)$ |
|---|---|---|---|
| t | t | t | 0.7 |
| t | t | f | 0.3 |
| t | f | t | 0.1 |
| t | f | f | 0.9 |
| f | t | t | 0.4 |
| f | t | f | 0.6 |
| f | f | t | 0.5 |
| f | f | f | 0.5 |

| $A$ | $B$ | $C$ | $f_C(ABC)$ |
|---|---|---|---|
| t | t | t | 0 |
| t | t | f | 1 |
| t | f | t | 1 |
| t | f | f | 0 |
| f | t | t | 1 |
| f | t | f | 0 |
| f | f | t | 0 |
| f | f | f | 1 |

(a) CPT for $C$                  (b) Mechanism for $C$

Figure 1: Two CPTs for variable $C$ with parents $A, B$. The second CPT represents a mechanism for variable $C$.

central to human reasoning [Pearl and Mackenzie, 2018] as we provide a formal account of how causal knowledge, even in this abstract form, can be quite useful computationally.

We start next with some further motivation, technical preliminaries and a review of the key results in [Darwiche, 2020]. We then study two key ingredients which are needed to formally define causal treewidth: jointree thinning and mechanism replication. We finally define causal treewidth and present some experimental results that shed more light on this notion and its underlying ingredients. Proofs of all results can be found in the appendix.

## 2 MOTIVATION AND PRELIMINARIES

Variables are discrete and denoted by uppercase letters (e.g., $X$) and their values are denoted by lowercase letters (e.g., $x$). Sets of variables are denoted by boldface, uppercase letters (e.g., $\mathbf{X}$) and their instantiations are denoted by boldface, lowercase letters (e.g., $\mathbf{x}$). A *factor* $f(\mathbf{X})$ is a mapping from instantiations $\mathbf{x}$ to non-negative numbers. A Bayesian network is a DAG $G$ together with one conditional probability table (CPT) for each node $X$ and its parents $\mathbf{P}$ in the DAG. A CPT specifies a conditional distribution $\Pr(X|\mathbf{P})$ and will be represented by a factor $f(X\mathbf{P})$ where $f(x\mathbf{p}) = \Pr(x|\mathbf{p})$ (hence, $\sum_x f(x\mathbf{p}) = 1$). To indicate that factor $f(X\mathbf{P})$ is a CPT for variable $X$, we will usually notate it as $f(X, \mathbf{P})$ or $f_X(X\mathbf{P})$. Of particular interest are CPTs (factors) that specify functions, also referred to as mechanisms.

**Definition 1.** *A factor $f(X, \mathbf{P})$ is a mechanism for $X$ (or $X$-mechanism) iff $f(x, \mathbf{p}) \in \{0, 1\}$ and $\sum_x f(x, \mathbf{p}) = 1$.*

A mechanism for variable $X$ represents a function whose inputs are parents $\mathbf{P}$ and whose output is $X$. Figure 1 depicts two factors over binary variables $\{A, B, C\}$. The factor in Figure 1a is a CPT for variable $C$ but is not a mechanism. The one in Figure 1b is also a CPT for $C$ but is a mechanism which corresponds to the function $C = A \oplus B$.

The use of mechanisms is ubiquitous in causality [Pearl, 2000]. In this context, root nodes in the DAG are called *exogenous* and internal nodes are called *endogenous*. A common class of models known as functional Bayesian net-

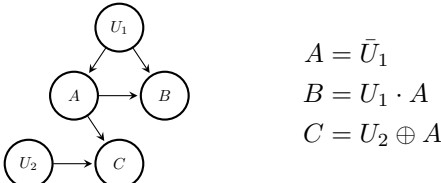

$$A = \bar{U}_1$$
$$B = U_1 \cdot A$$
$$C = U_2 \oplus A$$

Figure 2: SCM with endogenous variables $A, B, C$ and exogenous variables $U_1, U_2$. All variables are binary. The mechanisms for endogenous variables are specified by structural equations.

works or Structural Causal Models (SCMs) assume that the CPTs of all endogenous variables are mechanisms. Figure 2 depicts an example SCM where the mechanisms for endogenous variables $(A, B, C)$ are specified using structural equations as is commonly done. For this model to be complete, one also needs the CPTs for exogenous variables $(U_1, U_2)$ which specify the distributions $Pr(U_1)$ and $Pr(U_2)$, the only source of uncertainty in the model.

A classical setup in causal inference is to only have the graph of an SCM while assuming that the mechanisms (structural equations) are not known. In Figure 2, this would amount to assuming that each endogenous variable $(A, B, C)$ is a function of its parents, yet without knowing what these functions are. For example, we may not know whether the function for variable $C$ is $C = U_2 \oplus A$ or $C = U_2 + \bar{A}$ or $C = U_2 \cdot A$ or something else (there are 16 possible mechanisms for a binary variable with two binary parents). This situation may also arise in non-causality contexts where the assumption of unknown mechanisms can be viewed as background knowledge; see, e.g., [Chen et al., 2020].

In these situations, one typically has data in addition to the graph of a Bayesian network and the goal is to perform inference based on this available information; for example, by first estimating model parameters as suggested in [Zaffalon et al., 2021, Darwiche, 2021b, Chen et al., 2020]. This requires inference algorithms whose complexity is independent of the model parameters. Until relatively recently, the best complexity one could attain in this case is exponential in the graph treewidth. This complexity has been improved exponentially though due to the results in [Darwiche, 2020] and one goal of our work is to improve and further formalize these recent advances using the notion of causal treewidth.[2]

## 3   COMPUTING MARGINALS

There are two operations on factors, multiplication and sumout, which allow us to define the computational problem

whose complexity we wish to bound using causal treewidth. The *product* of factors $f(\mathbf{X})$ and $g(\mathbf{Y})$ is another factor $h(\mathbf{Z})$, where $\mathbf{Z} = \mathbf{X} \cup \mathbf{Y}$ and $h(\mathbf{z}) = f(\mathbf{x})g(\mathbf{y})$ for the unique instantiations $\mathbf{x}$ and $\mathbf{y}$ that are compatible with instantiation $\mathbf{z}$. *Summing-out* variables $\mathbf{Y} \subseteq \mathbf{X}$ from factor $f(\mathbf{X})$ yields another factor $g(\mathbf{Z})$, where $\mathbf{Z} = \mathbf{X} \setminus \mathbf{Y}$ and $g(\mathbf{z}) = \sum_{\mathbf{y}} f(\mathbf{yz})$. We will use $\sum_{\mathbf{Y}} f$ to denote the resulting factor $g$. We will also use $\sum_{\bar{\bar{\mathbf{Z}}}} f$ to denote summing out all variables from factor $f$ except for variables $\mathbf{Z}$. That is, for a factor $f(\mathbf{X})$, we will write $\sum_{\bar{\bar{\mathbf{Z}}}} f$ to mean $\sum_{\mathbf{Y}} f$ where $\mathbf{Y} = \mathbf{X} \setminus \mathbf{Z}$.

---
[2]A reviewer suggested using the term *functional treewidth* instead of *causal treewidth*. Our choice for the latter term is motivated by the emphasis we wish to place on exploiting "unknown" functions which are prevalent in causal inference, in contrast to the more common and informed exploitation of "known" functions.

The *joint distribution* of a Bayesian network is the product of its CPTs. The network on the right has CPTs $f_A(A)$, $f_B(AB)$, $f_C(AC)$, $f_D(BCD)$ and $f_E(CE)$. Its joint distribution is $\Pr(ABCDE) = f_A f_B f_C f_D f_E$. We can now compute the marginal over any variables by suming out all other variables from the joint distribution. For example, the marginal over variable $D$ is the factor $\Pr(D) = \sum_{ABCE} f_A f_B f_C f_D f_E = \sum_{\bar{\bar{D}}} f_A f_B f_C f_D f_E$. It is this computation of marginals that we will be bounding using causal treewidth. We are particularly interested in computing marginals over *families,* where a family is a variable and its parents, since these marginals form the basis of parameter estimation using algorithms such as gradient descent and EM; see, e.g., [Darwiche, 2009, Ch 17].

**Definition 2.** *Consider a DAG $G$ with nodes $X_1, \ldots, X_n$ and let $\mathbf{P}_i$ be the parents of $X_i$. Given a set of factors $f(X_i \mathbf{P}_i)$ for $i = 1, \ldots, n$, the marginals problem is to compute the factor $\sum_{\bar{\bar{\mathbf{F}}}} \prod_{i=1}^{n} f(X_i \mathbf{P}_i)$ for each family $\mathbf{F}$.*

A factor $f(X_i \mathbf{P}_i)$ will be called a *family factor.* The marginals problem does not place any restrictions on family factors so it is quite general. When these factors are CPTs, the marginals problem corresponds to the computation of marginals in a Bayesian network.

As mentioned earlier, if the Bayesian network has $n$ nodes and treewidth $w$, marginals can be computed in $O(n \cdot \exp(w))$ time and space. The simplest proof of this result is based on the algorithm of variable elimination (VE) which applies more generally to the problem in Definition 2 [Zhang and Poole, 1996, Dechter, 1996]. VE is based on two theorems, the first allows us to sum out variables in any order.

**Theorem 1.** $\sum_{\mathbf{XY}} f = \sum_{\mathbf{X}} \sum_{\mathbf{Y}} f = \sum_{\mathbf{Y}} \sum_{\mathbf{X}} f.$

The second theorem allows us to pull out factors from sums.

**Theorem 2.** *If variables $\mathbf{X}$ appear in factor $f$ but not in factor $g$, then $\sum_{\mathbf{X}} f \cdot g = g \sum_{\mathbf{X}} f.$*

Consider the factor $\sum_{ABDE} f(ACE)g(BCD)$. A direct computation of this factor multiplies factors $f$ and $g$ to yield the factor $h(ABCDE)$ and then sums out variables $ABDE$

from $g$. Using Theorem 1, we can arrange the above sum into $\sum_{AE} \sum_{BD} f(ACE)g(BCD)$. Using Theorem 2, we can arrange it further into $\sum_{AE} f(ACE) \sum_{BD} f(BCD)$. This is more efficient to compute as the largest factor constructed in the process will be over 3 instead of 5 variables.

Suppose we eliminate variables according to order $\pi$ when computing a marginal and let $w + 1$ be the largest number of variables appearing in a factor constructed in the process. The time and space complexity of VE can then be bounded by $O(n \cdot \exp(w))$ where $n$ is the number of variables. The number $w$ is called the *width* of order $\pi$. If the DAG has treewidth $w$ then there must exist an elimination order of width $w$. Moreover, no elimination order can have a width less than $w$; see [Darwiche, 2009, Ch 6 & 9] for a detailed exposition of these concepts and results.

## 4 EXPLOITING UNKNOWN MECHANISMS

Two new theorems were added to VE by Darwiche [2020] which enabled the exploitation of unknown causal mechanisms. In the following three results, we will use $\mathcal{F}, \mathcal{G}, \mathcal{H}$ to denote sets of factors, where each set is interpreted as a product of its factors. For example, the set of factors $\mathcal{F}$ will be interpreted as the factor $\prod_{f \in \mathcal{F}} f$.

**Theorem 3** ([Darwiche, 2020]). *Let $f$ be a mechanism for variable $X$. If $f \in \mathcal{G}$ and $f \in \mathcal{H}$, then $\mathcal{G} \cdot \mathcal{H} = \mathcal{G} \sum_X \mathcal{H}$.*

According to this result, if a mechanism for $X$ appears in both parts of a product, then variable $X$ can be summed out from one part without changing the value of the product.

**Corollary 1** ([Darwiche, 2020]). *If $f$ is a mechanism for $X$, $f \in \mathcal{G}$ and $f \in \mathcal{H}$, then $\sum_X \mathcal{G} \cdot \mathcal{H} = (\sum_X \mathcal{G})(\sum_X \mathcal{H})$.*

That is, if a mechanism for $X$ appears in both parts of a product, we can sum out variable $X$ from the product by independently summing it out from each part. Corollary 1 may appear unusable as it is predicated on multiple occurrences of a mechanism whereas the factors of a Bayesian network contain a single mechanism for each variable. This is where the second theorem comes in: *replicating* (i.e., duplicating) mechanisms in a product does not change the product value.

**Theorem 4** ([Darwiche, 2020]). *For mechanism $f$, if $f \in \mathcal{G}$, then $f \cdot \mathcal{G} = \mathcal{G}$.*

Consider the factor $\alpha = \sum_X f(XY)g(XZ)h(XW)$. VE has to multiply factors $f$, $g$ and $h$ before summing out variable $X$, therefore constructing a factor over four variables $XYZW$. However, if factor $f$ is a mechanism for variable $X$, then we can replicate it by Theorem 4: $\alpha = f(XY)g(XZ)f(XY)h(XW)$. Corollary 1 then gives $\alpha = \sum_X f(XY)g(XZ) \sum_X f(XY)h(XW)$. Hence, we can

---

**Algorithm 1** Complete Replication

1: **procedure** REPLICATE(DAG $G$, Functional nodes $\Gamma$ in $G$)
2: $\quad \Sigma \leftarrow$ multi-set of family factors of $G$
3: $\quad$ **for** each node $X$ in $\Gamma$ (bottom-up traversal) **do**
4: $\quad\quad$ **if** $X$ is a leaf **then** continue
5: $\quad\quad n \leftarrow$ number of $X$-feeding factors in $\Sigma$
6: $\quad\quad \Sigma \leftarrow \Sigma \cup \{n - 1 \text{ copies of the family factor for } X\}$
7: $\quad$ **return** $\Sigma$

---

now compute factor $\alpha$ without having to construct any factor over more than three variables. Moreover, we were able to do this without needing to know the function represented by factor $f(XY)$: we only needed to know that this factor represents a function from $Y$ to $X$. As shown in [Darwiche, 2020], this technique can lead to exponential savings that are attained without needing to know the identity of mechanisms which is a major departure from earlier techniques.

As the above example shows, the exploitation of unknown mechanisms requires their replication (duplication). A specific replication strategy was mentioned briefly and informally in [Darwiche, 2020] and referred to as a "heuristic." We shall call it the *complete replication strategy* for a reason that will become apparent later. This strategy is described formally in Algorithm 1 and uses the following definition.

**Definition 3.** *A family factor $f(X, \mathbf{P})$ is said to be $\underline{Y\text{-feeding}}$ iff $Y \in \mathbf{P}$.*

Algorithm 1 works with a *multi-set* of factors $\Sigma$ instead of a *set* since $\Sigma$ may contain multiple copies of the same factor. It starts with $\Sigma$ containing all family factors and traverses the DAG $G$ bottom up. When visiting a functional node $X$, it adds replicas of the mechanism for $X$ to $\Sigma$. Algorithm 1 returns what is called a *replication* of family factors.

**Definition 4.** *A $\underline{\text{replication of factors}}$ $\mathcal{F}$ is a multi-set $\mathcal{F}' \supseteq \mathcal{F}$ obtained by replicating some of the mechanisms in $\mathcal{F}$.*

Consider the DAG in Figure 3(a) where nodes $B$ and $C$ are functional. Calling Algorithm 1 on this DAG and these functional nodes returns the following replication $f_A(A)$, $f_B(AB)$, $f_B(AB)$, $f_B(AB)$, $f_C(BC)$, $f_C(BC)$, $f_D(BCD)$, $f_E(CE)$, which contains three replicas of the mechanism for $B$ and two replicas of the mechanism for $C$.

Even though a replication is technically a multi-set, we will simply refer to it as set for convenience. We will study (complete) mechanism replication extensively later.

A popular mechanization of VE is based on the notion of a *jointree*. We will review jointrees next as we shall use them to mechanize the exploitation of Theorems 3 and 4 and to formally define the notion of causal treewidth.

**Definition 5.** *A $\underline{\text{jointree}}$ for factors $\mathcal{F}$ is a tree in which every leaf node $i$ is assigned a non-empty set of factors $\mathcal{F}_i$*

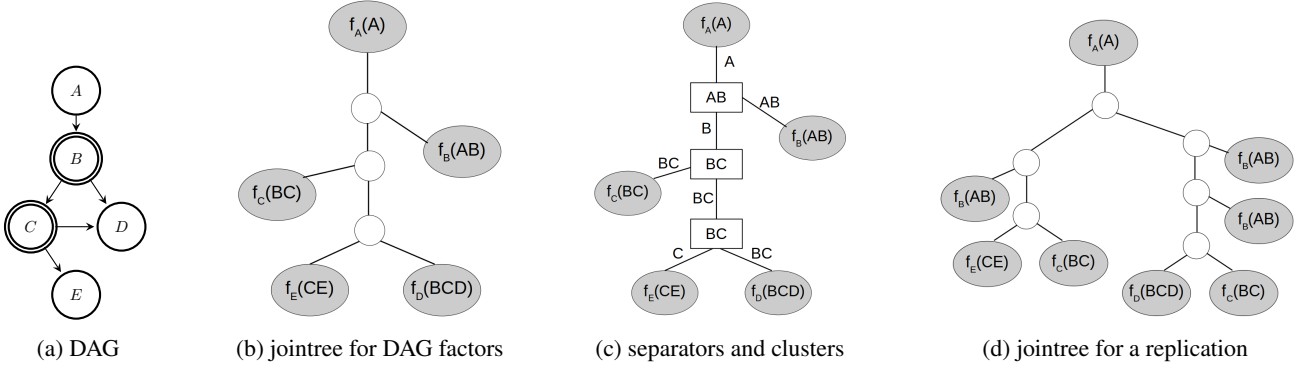

| (a) DAG | (b) jointree for DAG factors | (c) separators and clusters | (d) jointree for a replication |

Figure 3: A DAG with a jointree for its family factors (b,c) and a jointree for a replication of these factors (d).

*where the sets $\{\mathcal{F}_i\}_i$ form a partition of factors $\mathcal{F}$.* [3]

When a factor appears in $\mathcal{F}_i$, we will say that leaf node $i$ *hosts* the factor. We will use $\mathtt{vars}(i)$ to denote the variables of factors $\mathcal{F}_i$. For a jointree edge $(i, j)$, we will use $\mathtt{vars}(i, j)$ to denote the union of $\mathtt{vars}(k)$ for every leaf node $k$ on the $i$-side of the edge. Figure 3(b) depcits a jointree for the family factors of the DAG in Figure 3(a). Each leaf node of this jointree hosts exactly one factor.

A jointree induces edge and node labels as follows.

**Definition 6.** *The* separator $\mathbf{S}_{ij}$ *of jointree edge $(i, j)$ is defined as* $\mathtt{vars}(i, \overline{j}) \cap \mathtt{vars}(j, \overline{i})$. *If node $i$ is a leaf, its* cluster $\mathbf{C}_i$ *is defined as* $\mathtt{vars}(i)$, *otherwise as* $\bigcup_j \mathbf{S}_{ij}$. *The* width *of a jointree is the size of its largest cluster minus one.*

Figure 3(c) depicts the separators and clusters for the jointree in Figure 3(b). The width of this jointree is 2 since its largest cluster has 3 variables.

Jointrees play at least two key roles. First, their structure provides a specific recipe for when to multiply factors and when to sum out variables when applying VE. Second, their separators and clusters define the variables of factors constructed by VE so the sizes of these separators and clusters can be used to precisely determine the complexity of VE. We explain both roles next, starting with the following theorem which shows how a jointree can be used to direct VE towards the computation of marginals over separators.

**Theorem 5.** *Consider a jointree for factors $f_1, \dots, f_n$. Define the* message *from jointree node $i$ to its neighbor $j$ as:*

$$\mathcal{M}_{ij} = \begin{cases} \sum_{\overline{\overline{\mathbf{S}}}_{ij}} \mathcal{F}_i & \textit{for leaf node } i \\ \sum_{\overline{\overline{\mathbf{S}}}_{ij}} \prod_{k \neq j} \mathcal{M}_{ki} & \textit{for internal node } i \end{cases}$$

*For all jointree edges $(i, j)$, $\mathcal{M}_{ij} \mathcal{M}_{ji} = \sum_{\overline{\overline{\mathbf{S}}}_{ij}} f_1, \dots, f_n$.* [4]

Each message corresponds to a factor over some separator in the jointree. Hence, separators determine the space complexity of the message-passing algorithm of Theorem 5. A message $\mathcal{M}_{ij}$ can be computed in $O(\exp(|\mathbf{C}_i|))$ time and space given messages $\mathcal{M}_{ki}$ for $k \neq j$. Since $|\mathbf{C}_i| \leq w + 1$, where $w$ is the jointree width, all messages can be computed in $O(n \cdot \exp(w))$ time and space where $n$ is the number of jointree factors. Given an elimination order of width $w$, one can always construct a jointree of width $\leq w$; see [Darwiche, 2009, Ch 9]. Hence, the mechanization of VE using jointrees preserves the treewidth complexity bound.

In our context, jointrees play a third key role as they provide a direct method for exploiting Theorems 3 and 4 as shown in [Darwiche, 2020]. Instead of computing a jointree for the original set of factors $\mathcal{F}$, one computes a jointree for a replication $\mathcal{F}' \supseteq \mathcal{F}$ as licensed by Theorem 4; see Figure 3(d). One can then remove variables from separators and clusters in the expanded jointree based on Theorem 3 while preserving the soundness of the message passing algorithm. This reduces the jointree width and can lead to an exponential reduction in complexity. As in [Darwiche, 2021b], we refer to the process of removing variables from separators and clusters as the process of *thinning a jointree*.[5] We will show in the next section that the thinning procedure in [Darwiche, 2020] is not complete as it can miss opportunities that are licensed by Theorem 3. We will also provide a complete thinning procedure (with respect to Theorem 3) which paves the way for the formal definition of causal treewidth.

## 5 THINNING JOINTREES

Suppose we have a replication $\mathcal{F}' \supseteq \mathcal{F}$ of some factors $\mathcal{F}$. Given a jointree for the replication $\mathcal{F}'$, we will next define the notion of a *jointree thinning* and show that it is optimal

---

[3]Standard jointrees allow factors to be assigned to any node. Assigning factors to leaves, even one factor per leaf, does not preclude jointrees with optimal width; see [Darwiche, 2009, Ch 9].

[4]To compute the marginal over the family of variable $X$,

choose a leaf node $i$ in the jointree which hosts the family factor for $X$ and multiply this factor by message $\mathcal{M}_{ji}$ where $j$ is the single neighbor of $i$; see [Darwiche, 2009, Ch 7].

[5]The term "thin jointree" was used earlier in the context of approximate inference [Bach and Jordan, 2001].

(i.e., cannot be improved using Theorem 3). The replication $\mathcal{F}'$ may not be optimal though. Constructing optimal replications will be discussed in the next section.

**Definition 7.** *A jointree node $i$ is said to be X-connected to a factor $f$ iff $i$ hosts $f$ or $X$ appears in every separator on the path between $i$ and some leaf node $j$ that hosts factor $f$.*

**Definition 8.** *A jointree thinning maps every edge $(i,j)$ in the jointree to a set of variables $\mathbf{S}_{ij}^{\star} \subseteq \mathbf{S}_{ij}$, called a thinned separator, and satisfies two properties. First, for each functional variable $X \in \mathbf{S}_{ij}^{\star}$, we have:*

*(a) Node $i$ is not $X$-connected to any $X$-mechanism on the $i$-side of edge $(i,j)$, or node $j$ is not $X$-connected to any $X$-mechanism on the $j$-side of the edge.*

*(b) If node $i$ is not a leaf, then $X \in \mathbf{S}_{ik}^{\star}$ for some $k \neq j$.*

*(c) If node $j$ is not a leaf, then $X \in \mathbf{S}_{jk}^{\star}$ for some $k \neq i$.*

*Second, no other mapping from edges to supersets of $\mathbf{S}_{ij}^{\star}$ satisfies the above property.*

The separators $\mathbf{S}_{ij}$ of a jointree are determined by the locations of factors (the leaf nodes they are hosted at). Hence, the separators of a jointree are unique. However, a thinned separator $\mathbf{S}_{ij}^{\star}$ depends on both the locations of factors and other thinned separators. Hence, a jointree may have multiple thinnings. We define next the quality of a jointree thinning.

**Definition 9.** *A jointree thinning induces a thinned cluster $\mathbf{C}_i^{\star}$ for each jointree node $i$: If $i$ is a leaf, $\mathbf{C}_i^{\star} = \mathtt{vars}(\mathcal{F}_i)$; otherwise, $\mathbf{C}_i^{\star} = \bigcup_j \mathbf{S}_{ij}^{\star}$. The width of a jointree thinning is the size of its largest thinned cluster minus one.*

A jointree thinning leads to the notion of a causal jointree.

**Definition 10.** *A causal jointree is a jointree in which edges are annotated with thinned separators and nodes are annotated with thinned clusters. The causal width of a jointree is the smallest width attained by any of its causal jointrees.*

The width of a jointree can be determined by examining its cluster sizes. However, determining the causal width of a jointree is more involved as, in principle, it requires examining all thinnings of the jointree (causal jointrees).

**Theorem 6.** *The width of a jointree thinning and the causal width of a jointree are no greater than the jointree width.*

Figure 4 depicts a Bayesian network with two functional nodes $(B, C)$ and nine factors $\mathcal{F} = f_A, f_B, \ldots, f_I$. Consider now the replication $\mathcal{F}' \supseteq \mathcal{F}$ which results from duplicating mechanisms $f_B$ and $f_C$ once (that is, $\mathcal{F}'$ has 11 factors). Figure 5 depicts a jointree for the replication $\mathcal{F}'$ and two of its thinnings according to Definition 8. The one in Figure 5(a) has width 2. The one in Figure 5(b) has width 3.

Using thinned separators as given by Definition 8 preserves the correctness of the message passing algorithm.

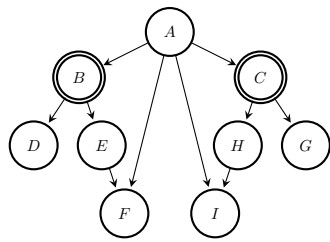

Figure 4: A Bayesian network with functional nodes $B$ and $C$.

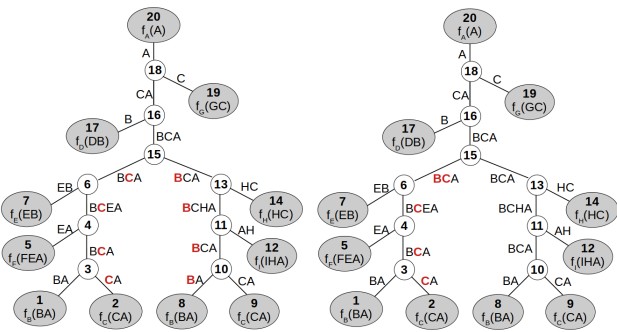

(a) jointree thinning of width 2    (b) jointree thinning of width 3

Figure 5: Two jointree thinnings. Each edge $(i, j)$ is marked by separator $\mathbf{S}_{ij}$. Red variables are not in the thinned separator $\mathbf{S}_{ij}^{\star}$.

**Theorem 7.** *Theorem 5 continues to hold if we use thinned separators $\mathbf{S}_{ij}^{\star}$ (as given by Definition 8) instead of classical separators $\mathbf{S}_{ij}$ (as given by Definition 6).*

The following result shows that we cannot improve Definition 8 of jointree thinnings based only on Theorem 3.

**Theorem 8.** *Consider a jointree thinning (Definition 8). If we remove any functional variable from a thinned separator, then Theorem 3 will no longer be sufficient to prove the soundness of the message-passing algorithm (Theorem 5).*

We next provide a characterization of jointree thinnings, which is more suitable for verifying whether the removal of variables from classical separators leads to a valid thinning.

**Theorem 9.** *A mapping from each jointree edge $(i, j)$ to variable set $\mathbf{S}_{ij}^{\star}$ is a jointree thinning according to Definition 8 iff (1) for each non-functional variable $X$, $X \in \mathbf{S}_{ij}^{\star}$ iff $X \in \mathtt{vars}(i, j) \cap \mathtt{vars}(j, i)$; (2) for each functional variable $X$: (a) if $X \in \mathtt{vars}(i)$ for a leaf node $i$, then $i$ is $X$-connected to exactly one mechanism for $X$; (b) if $X \in \mathbf{S}_{ij}^{\star}$ for some non-leaf $i$, then $X \in \mathbf{S}_{ik}^{\star}$ for some $k \neq j$.*

Definition 8 tells us what a thinning is but it does not tell us how to obtain one. We next provide a set of *thinning rules* that will generate every thinning admitted by Definition 8.

**Theorem 10.** *We can obtain a jointree thinning by starting with $\mathbf{S}_{ij}^{\star} = \mathbf{S}_{ij}$ and then removing variables from $\mathbf{S}_{ij}^{\star}$ according to the following rules, until no rules can be applied. Remove functional variable $X$ from $\mathbf{S}_{ij}^{\star}$ if either*

(a) *Node $i$ is $X$-connected to some $X$-mechanism on the $i$-side of edge $(i, j)$ and node $j$ is $X$-connected to some $X$-mechanism on the $j$-side of the edge; or*

(b) *$X \notin \mathbf{S}^\star_{ki}$ for all $k \neq j$ when node $i$ is not a leaf; or*

(c) *$X \notin \mathbf{S}^\star_{jk}$ for all $k \neq i$ when node $j$ is not a leaf.*

We will use $R_a(i, j, X)$ to mean that Rule (a) is applicable to variable $X$ and edge $(i, j)$ and call it a *rule application*. Similarly for $R_b(i, j, X)$ and $R_c(i, j, X)$. A jointree thinning can now be specified using a sequence of rule applications. The thinning in Figure 5(b) corresponds to $R_a(6, 15, C)$, $R_c(4, 6, C)$, $R_c(3, 4, C)$, $R_c(2, 3, C)$, $R_a(6, 15, B)$. The one in Figure 5(a) corresponds to $R_a(6, 15, C)$, $R_c(4, 6, C)$, $R_c(3, 4, C)$, $R_c(2, 3, C)$, $R_a(13, 15, B)$, $R_c(11, 13, B)$, $R_c(10, 11, B)$, $R_c(8, 10, B)$.

**Definition 11.** *A thinning sequence is a list of rule applications $R^1, \ldots, R^n$ where each rule is valid when it is applied and no rules are applicable after the sequence terminates.*

Theorem 10 says that the thinning rules are sound. The next result says they are complete (with respect to Definition 8).

**Theorem 11.** *Every causal jointree can be obtained using some thinning sequence.*

Two distinct thinning sequences may yield the same jointree thinning since the order of applying rules may not matter in some cases. The following result suggests a restriction on thinning sequences that does not compromise their ability to discover every possible jointree thinning.

**Theorem 12.** *Every jointree thinning can be obtained by a thinning sequence in which all applications of Rule (a) come before the applications of Rules (b,c).*

That is, we can first exhaust all applications of Rule (a) and then apply Rules (b,c). In fact, once we exhaust all applications of Rule (a), applying Rules (b,c) becomes deterministic. In other words, the jointree thinning obtained by a thinning sequence is fully determined by its Rules (a).

Thinning sequences mechanize the thinning process but finding an optimal thinning sequence remains a computationally challenging task given the large number of such sequences (even under the above restriction). Hence, one needs either sophisticated search algorithms or a heuristic to decide which thinning rule to apply and when. One heuristic that we found effective is to prefer $R_a(i, j, X)$ with the largest $\mathbf{S}^\star_{ij}$, followed by $X$ that is contained in the fewest neighboring separators, followed by minimizing the number of $X$-connected $X$-mechanisms on either side of edge $(i, j)$.

[Darwiche, 2020] proposed three thinning rules that apply only to binary jointrees in which each node has one or three neighbors [Shenoy, 1996]. The rules are not complete though as they can miss thinnings admitted by Definition 8. As in the rules we defined above, one starts by setting

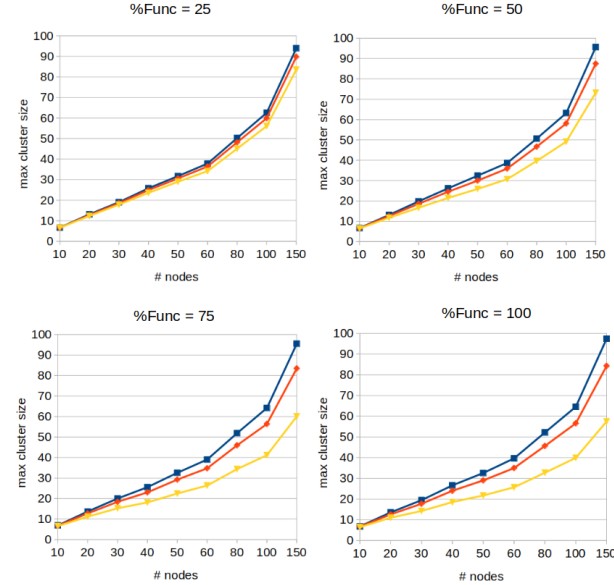

Figure 6: Comparing the thinning rules in Theorem 10 with the ones in [Darwiche, 2020]. Functional nodes are restricted to be internal (non-root) nodes. The average time for applying the new thinning rules to networks with 150 variables and 100% functional (hardest configuration) is 6.86 sec, with a min/max time of 0.8/28.8 sec.

thinned separators $\mathbf{S}^\star_{ij}$ to classical separators $\mathbf{S}_{ij}$ and then tries to remove variables from $\mathbf{S}^\star_{ij}$ using the rules. However, these rules can only be applied when visiting the jointree nodes in a particular order. A leaf node $h$ is identified first and then nodes are visited based on their distance from $h$, where the closer nodes are visited first. Suppose we are visiting a non-leaf node $i$. Let $p$ be its neighbor that is closest to leaf $h$ and let $c_1$ and $c_2$ be its two other neighbors. The first two rules require the following conditions: $X \in \mathbf{S}^\star_{ic_1}$, $X \in \mathbf{S}^\star_{ic_2}$, an $X$-mechanism is hosted on the $c_1$-side of edge $(c_1, i)$ and an $X$-mechanism is hosted on the $c_2$-side of edge $(c_2, i)$. If we further have $X \in \mathbf{S}^\star_{ip}$, the first rule licenses the removal of $X$ from either $\mathbf{S}^\star_{ic_1}$ or $\mathbf{S}^\star_{ic_2}$. If $X \notin \mathbf{S}^\star_{ip}$, the second rule licenses the removal of $X$ from both $\mathbf{S}^\star_{ic_1}$ and $\mathbf{S}^\star_{ic_2}$. The final rule applies to the single neighnor $r$ of leaf $h$, allowing us to remove variable $X$ from $\mathbf{S}^\star_{hr}$ when an $X$-mechanism is hosted at leaf $h$ and also at some other leaf in the jointree. The first rule involves a choice which is made using a heuristic described in [Darwiche, 2020].

Consider now the Bayesian network in Figure 4 and its thinned jointree in Figure 5(a) which has width 2. The best thinning that can be obtained by the rules in [Darwiche, 2020] has width 3, regardless of which leaf node $h$ we choose and regardless of what choices we make when applying the first rule. Figure 6 depicts a comparison between these rules and the ones in Theorem 10 on random (binary) jointrees, for the factors of complete replications generated by Algorithm 1. The plots in this figure vary the number of

Bayesian network nodes from 10 to 150 and consider different percentages of functional nodes (25, 50, 75 and 100) which are restricted to be non-root nodes.[6] They report the mean of maximal cluster size (width+1) over 10 jointrees for each data point. The plots are for the cluster sizes of (1) a classical jointree (blue), (2) a causal jointree obtained by the incomplete rules (red) and (3) a causal jointree obtained by the proposed rules (yellow). Four patterns are clear: more thinning takes place as we increase the number of functional nodes; the proposed thinning rules are much more effective; the gap between the two sets of rules grows as we increase the number of Bayesian network nodes and the number of functional nodes; the exploitation of unknown mechanisms can lead to significant reduction in inference complexity.

# 6 MECHANISM REPLICATION

The definition of thinning that we developed in the previous section was with respect to a particular replication $\mathcal{F}$ and a particular jointree for the factors in $\mathcal{F}$. Some replications are better than others in that they lead to causal jointrees of smaller width. We formalize this next.

**Definition 12.** *The width of a replication $\mathcal{F}$ is defined as the minimum width attained by any causal jointree for $\mathcal{F}$.*

Given a replication $\mathcal{F}$, we need to examine two search spaces before we can determine its width. First, we must choose a jointree for the factors in $\mathcal{F}$. Second, we must choose a particular thinning of the jointree. Hence, determining the width of a replication is not a straightforward task. Moreover, the width of a replication is not the only measure of its quality as we need to also consider its size. This is a critical issue that was not discussed in [Darwiche, 2020, 2021b] and that we need to explore carefully before we are ready to provide the formal definition of causal treewidth.

The size of a replication is the number of factors it contains (replicas are counted individually). To highlight the importance of a replication size, consider two replications $\mathcal{F}_1$ and $\mathcal{F}_2$ with respective sizes $n_1$ and $n_2$. Suppose now that replication $\mathcal{F}_1$ has width $w_1$ and replication $\mathcal{F}_2$ has width $w_2$. This means that there exists a causal jointree for replication $\mathcal{F}_1$ of width $w_1$ and no other causal jointree can have a smaller width (and similarly for replication $\mathcal{F}_2$). If we use these optimal causal jointrees, inference using these replications can be done in $O(n_1 \cdot \exp(w_1))$ and $O(n_2 \cdot \exp(w_2))$ time and space, respectively. One may be tempted to choose the replication with smaller width since complexity is exponential in width but linear in size. However, the size of

---

[6]Our method for generating a Bayesian network with nodes $X_1, \ldots, X_n$ assumes that each node has at most five parents. We visit nodes $X_i$ from $i = 1$ to $i = n$. When visiting node $X_i$, we randomly choose a number from $\{0, \ldots, \min(5, i-1)\}$ to represent the number of parents for $X_i$ and then randomly choose that many parents from $X_1, \ldots, X_{i-1}$.

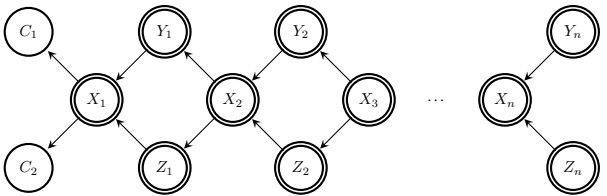

Figure 7: A DAG with an exponential complete replication.

a replication may also be exponential as we show next. In fact, the key result of this section is that the replication strategy proposed in [Darwiche, 2020], shown in Algorithm 1, satisfies two interesting properties. First, it is optimal: no other replication strategy will have a smaller width. Second, it can lead to replications of exponential size. We will in fact provide a bound on the size of replications produced by this strategy and suggest how it can be improved to avoid a blow up in replication size.

We start with the following result which shows that excessive replication can never hurt width.

**Theorem 13.** *Consider two replications $\mathcal{F}_1$ and $\mathcal{F}_2$ of some factors $\mathcal{F}$ where $\mathcal{F} \subseteq \mathcal{F}_1 \subseteq \mathcal{F}_2$. If the width of replication $\mathcal{F}_1$ is $w$, then the width of replication $\mathcal{F}_2$ is $\leq w$.*

We next identify a class of replications that possess some significant properties.

**Definition 13.** *A replication $\mathcal{F}$ is complete iff it satisfies the following property for each functional variable $X$ and its mechanism $f_X$. If $n$ is the number of $X$-feeding factors in $\mathcal{F}$ and $n > 0$, then $\mathcal{F}$ contains $n$ replicas of mechanism $f_X$. Otherwise, $\mathcal{F}$ contains only one replica of $f_X$.*

The first property of complete replications is uniqueness.

**Theorem 14.** *The family factors of any DAG have a unique complete replication. Moreover, the replications generated by Algorithm 1 are complete.*

The second property of complete replications is optimality.

**Theorem 15.** *Let $\mathcal{F}$ and $\mathcal{F}'$ be two replications of the family factors of a DAG. If replication $\mathcal{F}$ is complete, then its width is no greater than the width of replication $\mathcal{F}'$.*

The third property of complete replications is that their size can be exponential. Consider the family of DAGs in Figure 7 which have $3n + 2$ nodes for $n \geq 1$. Algorithm 1, which generates complete replications, starts with a set $\Sigma$ containing the $3n + 2$ family factors. It then visits functional nodes bottom-up and replicates their mechanisms. One can easily show that after visiting functional node $X_i$, the set $\Sigma$ will contain $2^i$ replicas of the mechanism for $X_i$.

We next provide a bound on the number of factors in a complete replication which suggests a method for controlling the potential blow up in its size.

**Definition 14.** *A functional chain of length $k$ in a DAG is a set of functional nodes $n_1, \ldots, n_k$ where node $n_i$ is a parent of node $n_{i+1}$ for $i = 1, \ldots, k-1$.*

**Theorem 16.** *Consider a DAG with $n$ nodes. Let $c$ be the largest number of children for any node and let $k$ be the length of longest functional chain. The (unique) complete replication of this DAG will contain at most $nc^k$ factors.*

This bound immediately provides a method for controlling the number of replicas in a complete replication. If we treat variable $X_{n/2}$ in Figure 7 as a non-functional variable, the length of the largest functional chain will be cut by half. Hence, by selectively ignoring some functional variables we can bound the size of functional chains and therefore ensure that Algorithm 1 will produce replications with size that is polynomial in the number of DAG nodes.

We are now ready to define causal treewidth formally.

**Definition 15.** *Consider a DAG $G$ with $n$ nodes, some of which are declared as functional. The causal treewidth of DAG $G$ is the smallest width attained by any replication $\mathcal{F}$ for $G$ where the size of $\mathcal{F}$ is polynomial in $n$.*

For the class of DAGs with bounded functional chains, one can use the complete replication $\mathcal{F}$ to determine the causal treewidth of the DAG. That is, determining causal treewidth becomes a matter of finding an optimal causal jointree for the factors in $\mathcal{F}$. The situation is more intricate for DAGs with unbounded functional chains. The complete replication cannot be used in this case and one must search among replications of polynomial size. It remains to be seen whether the space of replications to be explored can be restricted to subsets of the complete replication as suggested earlier. This is a subject of future work. We note here that [Darwiche, 2021b] identified a family of DAGs with $O(n^2)$ nodes, bounded depth and treewidth $n + 1$, while constructing thinned jointrees of width 2 for the family, assuming all internal nodes are functional. This is an example where the treewidth is unbounded while the causal treewidth is bounded, showing that causal treewidth dominates treewidth.

The appendix contains an experiment that reveals the importance of replication strategies and how such strategies interact with jointree construction methods. The experiment exhibited a number of patterns. First, the causal width was always smaller than the width, and quite substantially smaller, even when using random replications. Next, complete replications always produced a smaller causal width compared to random replications, particularly when the number of functional nodes is largest (100%). Finally, increasing the size of a random replication almost always correlated with decreasing the causal width but up to a certain point after which increasing the size of a replication did not help.

## 7 CONCLUSION

We provided a formal definition of the notion of causal treewidth, which dominates the classical and influential notion of treewidth. We also studied the three ingredients needed to define causal treewidth: mechanism replication, jointree construction and jointree thinning which yields causal jointrees. On the first front, we presented a number of results about a replication strategy that we called complete replication, showing that it is optimal while providing a bound on the size of replications it produces and suggesting a technique for controlling their size. On the second front, we highlighted the relevance (and irrelevance) of classical jointree construction methods to the construction of jointrees for replications. On the third front, we provided a complete characterization of causal jointrees and provided three thinning rules that are sound and complete for generating causal jointrees. We also proved some properties of these rules which can be of practical significance. We finally presented some experimental results to shed further light on the developments in this paper, which also showed that causal jointrees can lead to exponential improvements in the complexity of inference in comparison to jointrees.

### Acknowledgements

We wish to thank Yunqiu Han for useful feedback. This work has been partially supported by NSF grant #ISS-1910317 and ONR grant #N00014-18-1-2561.

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
