# OpenReview forum: "On the Definition and Computation of Causal Treewidth"
_auai.org/UAI/2022/Conference — UAI 2022 Poster_

### Official Review · Reviewer_epFw · 2022-03-25

**Q2(1) Originality/Novelty:** 1
**Q2(2) Significance/Impact:** 2
**Q2(3) Correctness/Technical Quality:** 3
**Q2(6) Clarity Of Writing:** 1
**Q6 Overall Score:** 3
**Q8 Confidence In Your Score:** 2

**Q1 Summary And Contributions:**

The paper is concerned with a notion of width
for Bayesian networks. I was going to say a "new notion", but it was
hard to tell if it is being claimed as being introduced here or whether
it was already present in some of its recent predecessor papers.

I really cannot say much more about the submission, for reasons explained further below.
But certainly it includes both a theoretical and experimental contribution.

**Q2 Assessment Of The Paper:**

More detailed information regarding each of these aspects is given below:

**Q2(4) Quality Of Experiments (Optional):**

3: Good: The experimental evaluation is adequate, and the results convincingly support the main claims.

**Q2(5) Reproducibility:**

3: Good: Key resources (e.g., proofs, code, data) are available and key details (e.g., proofs, experimental setup) are sufficiently well-described for competent researchers to confidently reproduce the main results.

**Q3 Main Strengths:**

The paper is on a core topic for UAI, certainly a bulls eye in terms of the scope of the conference.

The paper includes both theory and experiments

**Q4 Main Weakness:**

The paper is not only heavily based on [Darwiche 2020], but is chock full of
footnotes and asides mentioning [Darwiche 2020 and several others related
to it [Darwiche 2021].  Some of the comments mention differences/
deltas of the prior works, which will have little meaning to anyone who has not poured over
the prior works.

In addition to the issue of readability -- which to me was a show-stopper -- there is the question of the relationship of this paper to the prior ones.  Despite the constant references to the prior work, the paper lacked a lucid discussion of what is new.

The writing is surprisingly atrocious considering that the authors appear (from the paper's technical sophistication) to be experienced:

Some of the features that are particularly frustrating are:
-- numerous  forward references
-- cryptic and awkward phrasing (examples below)
-- incredibly compressed style; at times it seemed that the author
had taken a longer paper and just removed 4 out of 5 paragraphs


**Q5 Detailed Comments To The Authors:**

"We will show in the next section that the thinning procedure
in [Darwiche, 2020] is not complete though as it can miss
opportunities that are licensed by Theorem 3."

Thinning procedures have not been defined yet; and how useful
is it to know that a procedure you have not heard of is not complete.
And can an opportunity be licensed?

"Second, their
separators and clusters explicate the factors constructed by
VE"

How does a cluster explicate a factor?

"Our definition is optimal (i.e. no further thining can be done
while preserving the soundness of the message passing algorithm)"

The sentence, to the extent that it makes sense at all, is an exaggeration -- how does this show that a definition is optimal?

"We provided a formal definition of the notion of causal
treewidth, which dominates the classical and influential
notion of treewidth."

If dominates just means an inequality, then ok. But it appears
that some grander claim (and unjustified) might be indicated here.

**Q7 Justification For Your Score:**

My confidence is low in this area, and I can imagine that the paper
might make a significant contribution over its predecessors; certainly the authors appear to know the field inside out.  But even there is a lot to offer within it, I really do not think this version is ready for publication.

**Q9 Complying With Reviewing Instructions:**

1: Yes.

---

### Official Review · Reviewer_DFjH · 2022-04-11

**Q2(1) Originality/Novelty:** 2
**Q2(2) Significance/Impact:** 2
**Q2(3) Correctness/Technical Quality:** 3
**Q2(6) Clarity Of Writing:** 4
**Q6 Overall Score:** 6
**Q8 Confidence In Your Score:** 4

**Q1 Summary And Contributions:**

The paper builds on top of [Darwiche 2020] and proposes a more formal definition of causal treewidth, along with a couple algorithmic/heuristic results. Computation of (an upper bound for) causal treewidth involves three steps:

* CTa. Factor replication (this paper proposes a semi-trivial algorithm and proves it's optimality)

* CTb. Jointree construction (reuses [Darwiche 2020])

* CTc. Jointree thinning (proposes robust 'reduction rules' as well as a preliminary heuristic)


**Q2 Assessment Of The Paper:**

More detailed information regarding each of these aspects is given below:

**Q2(4) Quality Of Experiments (Optional):**

3: Good: The experimental evaluation is adequate, and the results convincingly support the main claims.

**Q2(5) Reproducibility:**

4: Excellent: Key resources (e.g., proofs, code, data) are available and key details (e.g., proof sketches, experimental setup) are comprehensively described for competent researchers to confidently and easily reproduce the main results.

**Q3 Main Strengths:**

The paper fixes some inadequacies in the definition of a promising width measure - causal treewidth. Albeit incremental, the contribution is fundamental and critical enough to impact the usage and adoption of the (more fine-grained) width measure in the future. Additionally, the paper also puts forth an algorithm for step (CTa) and proves its optimality.

Seems like a decent amount of effort went into the proofs, all of which unfortunately ended up in the appendix.


**Q4 Main Weakness:**

Overall incremental contribution. Apart from step (CTa) the contribution to the other two steps seems fairly minor and slightly unsatisfactory. Mainly, reusing previously proposed heuristics.
Eg. Page 7, "this is a subject of future work", what's addressed in the paper seems to be just the trivial case.


**Q5 Detailed Comments To The Authors:**

Experiment 2 doesn't seem to have a strong motivation and doesn't aid the paper much. Comparing the complete replication algorithm against the random replication strategy seems like a low threshold/easy baseline to pass. I would have prefered one of the proofs from the appendix in the main body instead (perhaps Thm 13?).

Appreciate the use of examples in the exposition.

Minor typos/grammar errors:

* [page 1, col2, para 2] 'which identities are unknown' unclear what this means

* [page 3, col2, para 2] 'single mechanism for each variable' should maybe be 'at most one mechanism for each variable'?

* [page 6, col2, para 1] 'neigh*n*or r of leaf'

* [page 7, col2, thm 14] 'will contain*s*'


**Q7 Justification For Your Score:**

In general, the parameter causal treewidth is quite new and seems promising. The paper would be critical in terms of the impact it can have on the future of causal treewidth because it proposes a rigorous/formal/robust definition which can act as a stable foundation. The provided experiments also vouch for the practical potential of causal treewidth and the proposed preliminary heuristics (for computing upper bounds on causal treewidth).


**Q9 Complying With Reviewing Instructions:**

1: Yes.

---

### Official Review · Reviewer_Vpeg · 2022-04-12

**Q2(1) Originality/Novelty:** 3
**Q2(2) Significance/Impact:** 2
**Q2(3) Correctness/Technical Quality:** 2
**Q2(6) Clarity Of Writing:** 4
**Q6 Overall Score:** 6
**Q8 Confidence In Your Score:** 3

**Q1 Summary And Contributions:**

The motivation is to improve the performance of marginal inference in Bayesian networks that contain functional nodes that depend deterministically on their parents.

They show that the factors of functional nodes can be replicated in the marginal without changing the result, and that these replications lead to a lower tree width.

They perform some experiments comparing random and complete replication.

**Q2 Assessment Of The Paper:**

More detailed information regarding each of these aspects is given below:

**Q2(4) Quality Of Experiments (Optional):**

3: Good: The experimental evaluation is adequate, and the results convincingly support the main claims.

**Q2(5) Reproducibility:**

3: Good: Key resources (e.g., proofs, code, data) are available and key details (e.g., proofs, experimental setup) are sufficiently well-described for competent researchers to confidently reproduce the main results.

**Q3 Main Strengths:**


Inference of Bayesian networks is an important problem and they show how to make it faster .

All their results are proven

**Q4 Main Weakness:**

I doubt that functional nodes are very common in practice. Deterministic functions run counterary to the purpose of having Bayesian inference

They mostly extend results of ([Darwiche, 2020]). The main algorithm was already known there

**Q5 Detailed Comments To The Authors:**

"Causal treewidth" sounds pretentious. Considering that it is about functional nodes, is it not more of a "functional treewidth"?

>p4 depicts a jointree for 11 factors, some of which are replicas.

Which ones are replica?

>Definition 6

"Shared" means the intersection? What are the factors of the j-side when the j side is an inner node (not a leaf node) ?
If the factors of all inner nodes are the empty set, then all S_ij are also the empty set? Then all clusters are also the empty set.
Nothing in the the definition says it cannot be the empty set. Is there something missing from the definition?

>Definition 15 Consider a DAG G with n nodes ... any replication F for G where the size of F is polynomial in n.

This restriction on the size of F is meaningless if the definition considers a certain given DAG. For any size s, you can always find a constant c, such that s <= n^c is polynomial.

Such a restriction only works for an infinite family of replications (e.g. generated by an algorithm) where you have a replication F(n) for each possible n where you can give an upper bound on the size of F(n) when n grows arbitrarily high.

>Figure 5

The labels could be improved. "cls size"  means cluster size? 100%_causal_width means 100% of nodes are functional? That should be mentioned in the caption

When 100% of the nodes are functional, do you even need a jointree? There is one assignment of values to the variables where the joint probability is 1, and all the other assignments give 0, so one just needs to evaluate the graph probability function once, which would give much better performance than the jointree.

Is there code for the experiments?


>p12 We expand message Mzk as follows:

There are closing parentheses missing in the following equation

**Q7 Justification For Your Score:**

I do not think the studied problem is very important, so the paper will probably not have a high impact

update: raised Reproducibility and final score, since they want to make code available

**Q9 Complying With Reviewing Instructions:**

1: Yes.

---

### Official Review · Reviewer_e9dk · 2022-04-18

**Q2(1) Originality/Novelty:** 2
**Q2(2) Significance/Impact:** 2
**Q2(3) Correctness/Technical Quality:** 3
**Q2(6) Clarity Of Writing:** 3
**Q6 Overall Score:** 4
**Q8 Confidence In Your Score:** 3

**Q1 Summary And Contributions:**


This paper considers causal treewidth.  It revisits definitions by (Darxiche 2020) and proposes an improvement to get more optimal thinnings of the jointrees.


**Q2 Assessment Of The Paper:**

More detailed information regarding each of these aspects is given below:

**Q2(4) Quality Of Experiments (Optional):**

2: Fair: The experimental evaluation is weak: important baselines are missing, or the results do not adequately support the main claims.

**Q2(5) Reproducibility:**

2: Fair: Key resources (e.g., proofs, code, data) are unavailable but key details (e.g., proof sketches, experimental setup) are sufficiently well-described for an expert to confidently reproduce the main results.

**Q3 Main Strengths:**




The problem is relevant and interesting.

The text is well written and mostly understandable.

An improvement is made over earlier work (Darwiche 2020)



**Q4 Main Weakness:**


Being more optimal in finding causal treewidth should be put in sufficient context.  Normal treewidth is NP-hard to determine, and in the submission there doesn't seem to be a strong complexity result (to the contrary, e.g., Theorem 14 has an exponential bound on the number of factors), so I suspect that even though the improvement allows with unbounded resources to find more often a better treewidth, it remains unclear how much computation this requires.

The text is quite abstract, it is hard to understand what exactly is assumed to be fully known (or could be unknown) as input for the considered problem.  It is unclear to me how this theory would be used practically when learning a Bayesian network or performing another canonical task.

The experiments are illustrative but don't aim to be reproducible.


**Q5 Detailed Comments To The Authors:**


Please specify more clearly when a causal mechanism is "unknown".

It seems important to put the term "causal" into context, since definitions and operations seem to just take a Bayesian network as input, without making strong assumptions about the direction of causality.

Theorem 3 gives a property of the product of two sets of factors.  It is desirable to first define this product, especially when the two factors contain common variables, so that expressions such as \mathcal{G} \sum_X \mathcal{H} can be correctly interpreted if boath \mathcal{G} and \mathcal{H} contain X.
There are other terms which aren't formally defined, e.g., replica.

In definition 3 & algorithm 1, in "X-feeding" on the one hand X seems to be a variable (e.g., in the expression "a mechanism for X" in alg 1 line 5, on the other hand X seems to be a node, e.g., in the expression "node X in G", or X may be a factor, e.g., in the expression Y \in \mathbf{P} in def 3 where P is a set of factors.

Definition 7 (and others) use terms as "node" which may both refer to the join tree and the bayesian network.  Often the meaning can be derived from the context but avoiding ambiguity risks greatly improves readability.

Theorem 14: will contains -> will contain



**Q7 Justification For Your Score:**

The paper seems interesting and sound, but has too many minor issues to be really convincing.

**Q9 Complying With Reviewing Instructions:**

1: Yes.

---

### Decision · Program_Chairs · 2022-05-15

**Decision:**

Accept (Poster)

**Comment:**

Meta Review: I think this paper is important, and that, as some reviewers point out, not an easy read. Probably the authors (and most of us do) were in a rush to submit the paper and neglected a certain care in making the paper somewhat more accessible. They also somewhat did not properly highlight the originality compared to previous work, notably by Darwiche. I think the discussion has been useful to clarify this point.

Anyway, the paper refines the notion of treewidth when there are functional nodes in a Bayesian net; since this is essentially the standard situation in structural causal models (uncertainty only in the roots, determinism everywhere else), the authors correctly - in my view - call the resulting notion causal treewidth. The causal treewidth can reduce the treewidth we would get without functional nodes, and hence jointree computational complexity.

One question to the authors, which could perhaps clarify part of the intuition in a revised version of the paper: saying that we can compute/define the causal treewidth even without knowing the function is a functional node, is it equivalent to consider the "worst-case" function among all the possible ones?

More generally speaking, even considered the shortcomings of the paper, I believe this paper should be accepted as it is serious and hard work. The authors should just make more of an effort to make the results much more accessible to a general audience.